# The European Badger as a New Host for *Dirofilaria immitis* and an Update on the Distribution of the Heartworm in Wild Carnivores from Romania

**DOI:** 10.3390/pathogens11040420

**Published:** 2022-03-30

**Authors:** Angela Monica Ionică, Georgiana Deak, Radu Boncea, Călin Mircea Gherman, Andrei Daniel Mihalca

**Affiliations:** 1Department of Parasitology and Parasitic Diseases, Faculty of Veterinary Medicine, University of Agricultural Sciences and Veterinary Medicine of Cluj-Napoca, 400372 Cluj-Napoca, Romania; ionica.angela@usamvcluj.ro (A.M.I.); calin.gherman@usamvcluj.ro (C.M.G.); amihalca@usamvcluj.ro (A.D.M.); 2Molecular Diagnosis Laboratory, Clinical Hospital of Infectious Diseases of Cluj-Napoca, 400348 Cluj-Napoca, Romania; 3Falcon Vet Veterinary Private Clinic, 021919 Bucharest, Romania; radu.boncea@yahoo.com

**Keywords:** *Dirofilaria immitis*, hosts, *Meles meles*, Romania, wild carnivores

## Abstract

*Dirofilaria immitis* is an important mosquito-borne nematode, being of both veterinary and public health concern. The typical final host is represented by the domestic dog, *Canis familiaris*, but it is able to infect a large variety of mammalian species. During the present study (March 2016–February 2022), a total of 459 wild carnivore carcasses belonging to 17 species, from Romania, were evaluated for the presence of adult *D. immitis* by necropsy. Overall, 20 animals (4.36%) were positive: twelve golden jackals, *C. aureus* (19.05%); four red foxes, *Vulpes vulpes* (6.67%); one raccoon dog, *Nyctereutes procyonoides*; two wild cats, *Felis silvestris* (4.65%); and one European badger, *Meles meles* (0.87%). This study provides further evidence of the occurrence of the canine heartworm, *D. immitis*, in Romania, expanding the known host spectrum, reports a new host species for this parasite, the European badger, and a new host for Europe, the raccoon dog.

## 1. Introduction

Among mosquito-borne nematodes, the genus *Dirofilaria* (Spirurida, Onchocercidae) is important from a veterinary and public health perspective. The two main species, *D. immitis* and *D. repens*, causative agents of canine heartworm disease and subcutaneous dirofilariosis, respectively, have a wide distribution and are able to infect a large variety of domestic and wild carnivorous host species [1,2]. Furthermore, *D. immitis* is also of major veterinary concern, as it is associated with a debilitating and eventually fatal disease in infected carnivores [1]. Although several species of mosquitoes, mainly within the genera *Culex*, *Aedes*, *Ochlerotatus*, and *Anopheles* may successfully transmit *D. immitis*, the main natural vectors are represented by *Aedes vexans*, *A. albopictus*, and the *Culex pipiens* complex [3,4].

The most frequently infected final host and the most competent reservoir is the domestic dog, *Canis familiaris*, but the infection was reported in over other 30 mammalian species, including several wild canids, domestic and wild felids, mustelids, ursids, ailurids, pinnipeds, monkeys, rodents, and ungulates [5,6,7]. However, the occurrence of patent infections in hosts other than dogs has rarely been documented. Felids, particularly domestic cats, are susceptible, but inappropriate reservoirs, as indicated by the low number of adult worms and absence or short duration of microfilariaemia [5,8]. Conversely, some species of wild canids, such as the golden jackals (*Canis aureus*), and grey wolves (*C. lupus*), seem to have an important epidemiological role in the maintenance and transmission of this parasite [2].

Romania is endemic for canine cardiac dirofilariosis, with *D. immitis* detected in dogs originating from most regions of the country [9,10,11]. Furthermore, *Dirofilaria* spp. infections were detected by necropsy and/or molecular tools also in six species of wild carnivores throughout the country [12,13].

The aims of the present study were to provide an update on the distribution of the heartworm, *D. immitis*, in wild carnivores in Romania, to report new host-parasite associations and evaluate their potential reservoir role.

## 2. Results

Overall, 20 out of the 459 (4.36%) examined animals harboured at least one adult *D. immitis* in the right ventricle or the pulmonary artery, most of which originated from the southern region of the country (Table 1, Figure 1).

Among canids, the overall prevalence of infection was 12.88% (95% CI 7.68–19.82), with the highest value recorded for golden jackals, *Canis aureus*, followed by red foxes, *Vulpes vulpes*. A single raccoon dog, *Nyctereutes procyonoides*, was examined and the animal was found to be positive. Within the felids, two wild cats, *Felis silvestris* (4.65%; 95% CI 0.57–15.81), were positive. Of all mustelids, one European badger, *Meles meles* (0.87%; 95% CI 0.02–4.75), was infected.

The worm burden ranged between one and seven nematodes/animal, with an average value of 2.35, and a median of 1. The individual details of all the infected animals are presented in Table 2.

## 3. Discussion

The present study represents an extension of previous work [12,13], and provides data obtained during further monitoring of heartworm infection in the Romanian wildlife. Most positive animals originated from the southern and south-eastern part of the country, where endemicity in domestic dogs’ populations has been repeatedly confirmed and the prevalence of infection was up to 26% [9,10].

A relatively high prevalence of infection was found in golden jackals, *C. aureus* (19.05%). This value is similar to the ones reported from our previous studies [12,13], but generally higher than most reports from other neighbouring countries: 7.32% in Serbia [14], 7.4% in Hungary [15], and 4.4–37.54% in Bulgaria [16,17,18]. Furthermore, seven out of the twelve positive animals harboured both male and female nematodes, which further suggests the involvement of this species as reservoir host. The actual frequency of microfilariaemia seems to be, in fact, lower, with only three individuals positive by means of microscopy (Figure 2a) and/or PCR. However, considering the adaptation used for the modified Knott’s test, we regard the microscopical negativity as questionable, while PCR positivity serves as indirect evidence of the presence of microfilariae. In contrast, although the relative prevalence of infection in red foxes, *V. vulpes*, was also high (6.67%), only one out of the four positive individuals harboured nematodes of both sexes, and was positive also for microfilariae (Figure 2b). The foxes are receptive hosts, with reported infection rates ranging between 0.4% and 25.22% in Europe [18,19]. However, the few studies where patency was evaluated seem to indicate a far lower frequency of patent infections as compared to the presence of adult nematodes [15,20,21].

A single raccoon dog, *N. procyonoides*, was examined throughout the study period. The animal harboured a single adult female of *D. immitis*. The raccoon dogs are introduced species in Europe, including Romania, and can be found mainly in wet habitats [22]. *Dirofilaria immitis* infections have been documented in raccoon dogs in Asia [23,24], but to the best of our knowledge, this represents the first report in Europe. In living animals, a seroprevalence of 7.4% was reported in raccoon dogs in Japan [23]. In Korea, the seroprevalence of investigated living individuals was of 17.8%, but microfilariae were not detected in any of the positive animals [24]. Raccoon dogs are not regarded as relevant reservoir hosts, as experimental infections have shown that the worm burden is low, and microfilariaemia lasts for a short period, of around 120 days [25]. 

Data regarding heartworm infection in wild felids is still scarce. Out of 48 examined carcasses (43 wild cats, *F. silvestris*, and 5 lynxes, *L. lynx*), 2 wild cats were positive, harbouring 1 adult male nematode each. Both cases have been published separately [26,27]. The presence of a single nematode further suggests the lack of reservoir competence for this species.

Among mustelids, *D. immitis* infection is known to occur in ferrets, *Mustela putorius furo*, which also act as reservoirs [28]. Occasionally, the parasite has been reported also in Eurasian otters, *Lutra lutra* [13,29,30], including one case with confirmed patency [31]. During the present study, we examined a total of 237 mustelid carcasses. Among them, a single European badger, *M. meles*, out of the 115 collected was positive. Two adult male and one female nematode were recovered. To the best of our knowledge, this represents the first record of *D. immitis* infection in this host species. Whether this was an accidental infection or the badger is, in fact, a suitable definitive host is unclear; however, this finding further expands the global list of known receptive host species. The occurrence of microfilariaemia was demonstrated both by microscopical examination (Figure 2c,d) and molecular detection, which indicates at least a temporary availability of microfilariae, and therefore, potential reservoir status. The duration and intensity of microfilariaemia require further investigations.

## 4. Materials and Methods

Between March 2016 and February 2022, a total of 459 wild carnivore carcasses belonging to 17 species were examined by parasitological necropsy (Table 3). The animals were either legally hunted or found dead as road-kills at various locations throughout the country and were stored at −20 °C until processing. For each examined animal, the species, sex, and geographical location were recorded. Whenever possible, the age of the animal was estimated according to dentition (juvenile or adult). The heart and pulmonary arteries were longitudinally dissected in order to assess the presence of adult *D. immitis*. All filarioids were collected in 70% ethanol, in labelled tubes, and morphologically identified under a dissection microscope, based on descriptions and keys available in the literature [32,33].

In animals that were positive by necropsy, despite no liquid blood being available, an attempt to assess the occurrence of microfilaraemia directly was made, by applying a modified Knott’s test [34] to coagulated blood retrieved from the heart. Genomic DNA was also isolated from blood clots of the positive individuals, and screened for filarial DNA, using the “panfilaria” PCR primers and protocol, as previously described [35].

The statistical analyses (prevalence and 95% Confidence Intervals) were performed using EpiInfo 7 software (version 7.2, CDC, Atlanta, GA, USA) and the map was generated using QGIS 3.4 software.

## 5. Conclusions

The present study provides further evidence of the occurrence of the canine heartworm, *Dirofilaria immitis*, in Romania, expanding the known host spectrum of this parasite. We report a new host for Europe, the raccoon dog, *Nyctereutes procyonoides*, and a new host-parasite association for the European badger, *Meles meles*.

## Figures and Tables

**Figure 1 pathogens-11-00420-f001:**
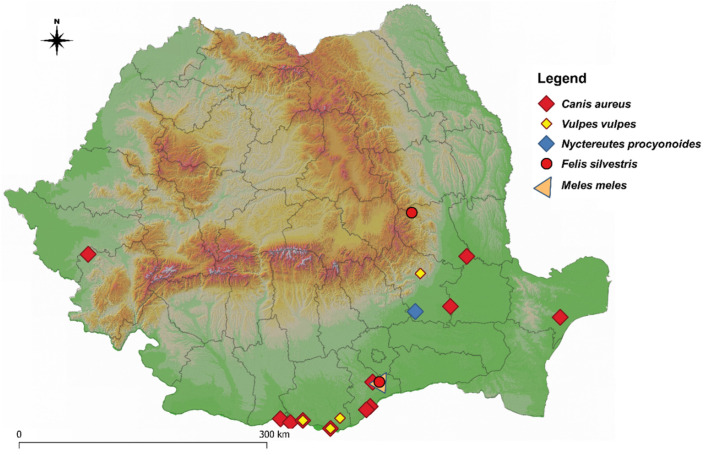
The distribution of *D. immitis* infection in wild carnivores examined in Romania.

**Figure 2 pathogens-11-00420-f002:**
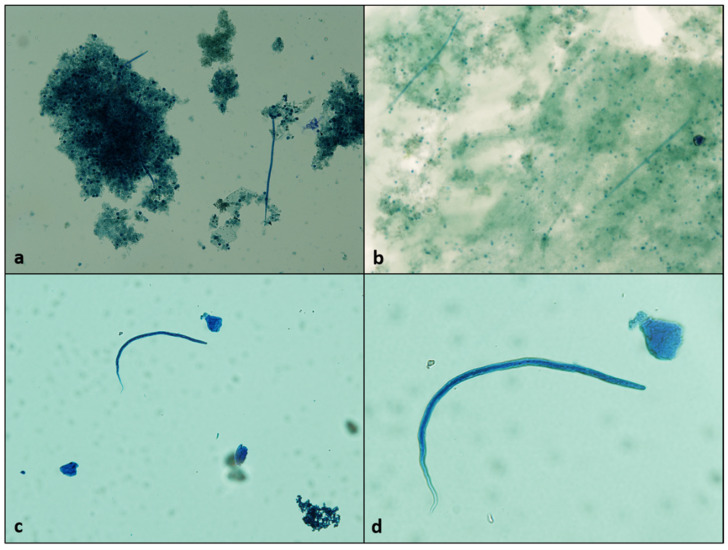
Microfilariae detected by modified Knott’s test in golden jackal (**a**), red fox (**b**), and European badger (**c**), 4× objective; and detail in European badger (**d**), 10× objective.

**Table 1 pathogens-11-00420-t001:** The prevalence of *D. immitis* infection in wild carnivores from Romania.

Family	Species	Examined	*Dirofilaria immitis*		
			n	%	95% CI
**Canidae**	*Canis aureus*	63	12	19.05	10.25–30.91
	*Canis lupus*	8	0	0	0–36.94
	*Nyctereutes procyonoides*	1	1	100	2.50–100
	*Vulpes vulpes*	60	4	6.67	1.85–16.20
	**Total**	**132**	**17**	**12.88**	**7.68–19.82**
**Felidae**	*Felis silvestris*	43	2	4.65	0.57–15.81
	*Lynx lynx*	5	0	0	0–52.18
	**Total**	**48**	**2**	**4.17**	**0.51–14.25**
**Mustelidae**	*Meles meles*	115	1	0.87	0.02–4.75
	*Mustela putorius*	80	0	0	0–4.51
	*Martes foina*	36	0	0	0–9.74
	*Lutra lutra*	21	0	0	0–16.11
	*Martes martes*	5	0	0	0–52.18
	*Mustela nivalis*	3	0	0	0–70.76
	*Mustela erminea*	1	0	0	0–97.50
	*Mustela lutreola*	1	0	0	0–97.50
	*Mustela eversmanii*	1	0	0	0–97.50
	*Vormela peregusna*	1	0	0	0–97.50
	**Total**	**264**	**1**	**0.38**	**0.01–2.09**
**Ursidae**	*Ursus arctos*	15	0	0	0–21.80
**Total**		**459**	**20**	**4.36**	**2.84–6.63**

**Table 2 pathogens-11-00420-t002:** *Dirofilaria immitis* infection in wild carnivores from Romania: overall results.

No.	Species	Region	Sex	Age	*Dirofilaria immitis*			
					F	M	Microfilariae	PCR
1	*Canis aureus*	West	M	Juvenile	1	1	Negative	Negative
2	*Canis aureus*	South	F	Adult	1	2	Positive	Positive
3	*Canis aureus*	South	F	Adult	1	2	Negative	Positive
4	*Canis aureus*	Southeast	M	Adult	2	1	Negative	Negative
5	*Canis aureus*	South	F	Adult	2	1	Negative	Negative
6	*Canis aureus*	South	M	Adult	4	1	Negative	Negative
7	*Canis aureus*	South	F	Juvenile	6	1	Negative	Positive
8	*Canis aureus*	Southwest	F	Adult	1	0	Negative	Negative
9	*Canis aureus*	South	M	Adult	1	0	Negative	Negative
10	*Canis aureus*	Southeast	M	not recorded	5	0	Negative	Negative
11	*Canis aureus*	South	M	Adult	0	1	Negative	Negative
12	*Canis aureus*	Southeast	M	Juvenile	0	1	Negative	Negative
13	*Vulpes vulpes*	South	M	Adult	1	1	Positive	Positive
14	*Vulpes vulpes*	South	F	Juvenile	2	0	Negative	Negative
15	*Vulpes vulpes*	South	F	not recorded	0	1	Negative	Negative
16	*Vulpes vulpes*	South	M	Juvenile	0	1	Negative	Negative
17	*Nyctereutes procyonoides*	Southeast	F	Adult	1	0	Negative	Negative
18	*Felis silvestris*	Southeast	M	Adult	0	1	Negative	Negative
19	*Felis silvestris*	South	F	Adult	0	1	Negative	Negative
20	*Meles meles*	South	M	Adult	1	2	Positive	Positive

**Table 3 pathogens-11-00420-t003:** Origin of examined wild carnivores from Romania.

Species	Examined	Region						
		SE	S	SW	W	NW	C	NE
**Canidae**								
*Canis aureus*	63	28	19	11	3	1	1	-
*Canis lupus*	8	-	-	-	2	1	5	-
*Nyctereutes procyonoides*	1	1	-	-	-	-	-	-
*Vulpes vulpes*	60	8	42	-	2	6	2	-
**Felidae**								
*Felis silvestris*	43	8	2	-	5	22	5	1
*Lynx lynx*	5	-	-	1	-	3	1	-
**Mustelidae**								
*Meles meles*	115	3	6	1	12	75	18	-
*Mustela putorius*	80	16	60	1	-	1	2	-
*Martes foina*	36	9	4	1	-	20	1	1
*Lutra lutra*	21	7	7	-	3	1	1	2
*Martes martes*	5	-	-	-	-	5	-	-
*Mustela nivalis*	3	-	1	-	1	1	-	-
*Mustela erminea*	1	-	-	-	-	1	-	-
*Mustela lutreola*	1	1	-	-	-	-	-	-
*Mustela eversmanii*	1	-	1	-	-	-	-	-
*Vormela peregusna*	1	1	-	-	-	-	-	-
**Ursidae**								
*Ursus arctos*	15	-	-	-	-	2	13	-
Total	459	82	142	15	28	139	49	4

SE: Southeast; S: South; SW: Southwest; W: West; NW: Northwest; C: Centre; NE: Northeast.

## Data Availability

The complete dataset used and analysed during the current study are available from the corresponding author on reasonable request.

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
