# Peer review of "The European Badger as a New Host for Dirofilaria immitis and an Update on the Distribution of the Heartworm in Wild Carnivores from Romania"

_pathogens, 2022, doi:10.3390/pathogens11040420_

Round 1
Reviewer 1 Report
In their manuscript "The European badger as a new host for Dirofilaria immitis and an update on the distribution of the heartworm in wild carnivores from Romania" the authors present the results of a 6-year survey of D. immitis in potential Romanian wildlife hosts. In addition to updating species infection rates, the authors demonstrate for the first time D. immitis infection of the European badger. The raccoon dog is also demonstrated as a susceptible reservoir for the first time in a European setting.
While the number of specimens of certain species are understandably low, reasonably large numbers of common potential hosts were obtained for several species. The detection of microfilariae cannot be as reliable as in freshly sampled blood, but the authors' methods supplemented by panfilarial PCR are adequate for this purpose. Data presentation is clear and the authors' claims are well supported. Some minor text editing for grammar is recommended.
As it is currently formatted, in table 1 the third column title ("Examined") wraps to a second line making it looking like "d" is an abbreviation for something.
Author Response
Dear reviewer 1, thank you for your kind words and for taking the time to review our paper. We appreciate your efforts. We corrected Table 1.
Reviewer 2 Report
Manuscript presents interesting data on the occurrence of Dirofilaria immitis (canine heartworm) in wild carnivores (including Canidae, Felidae, Mustelidae and Ursidae) in Romania. Presented study provides an update on the distribution of D. immitis in wild carnivores to report new host-parasite associations and evaluate their potential reservoir role. The study is well designed. The Authors examined as many as 459 wild carnivore carcasses. I can see the huge work done by the Authors related to the parasitological necropsy. The results are correctly presented and the manuscript is well written.
Author Response
Thank you very much for your time in revising our paper.